# Comprehensive RNA Expression Analysis Revealed Biological Functions of Key Gene Sets and Identified Disease-Associated Cell Types Involved in Rat Traumatic Brain Injury

**DOI:** 10.3390/jcm11123437

**Published:** 2022-06-15

**Authors:** Qilin Tang, Mengmeng Song, Rongrong Zhao, Xiao Han, Lin Deng, Hao Xue, Weiguo Li, Gang Li

**Affiliations:** 1Department of Neurosurgery, Qilu Hospital, Cheeloo College of Medicine and Institute of Brain and Brain-Inspired Science, Shandong University, Jinan 250012, China; qilin.tang@hotmail.com (Q.T.); qlyyzrr@163.com (R.Z.); tahanpeng@163.com (X.H.); denglinqilu@163.com (L.D.); xuehao@sdu.edu.cn (H.X.); leeweiguo777@163.com (W.L.); 2Shandong Key Laboratory of Brain Function Remodeling, Jinan 250012, China; 3Department of Nuclear Medicine, Qilu Hospital, Shandong University, Jinan 250012, China; mengmeng.song@outlook.com; 4Department of Neurosurgery, Children’s Hospital Affiliated to Shandong University, Jinan 250012, China

**Keywords:** traumatic brain injury, WGCNA, inflammatory response, microglia, perivascular macrophages

## Abstract

Traumatic brain injury (TBI) is a worldwide public health concern without major therapeutic breakthroughs over the past decades. Developing effective treatment options and improving the prognosis of TBI depends on a better understanding of the mechanisms underlying TBI. This study performed a comprehensive analysis of 15 RNA expression datasets of rat TBIs from the GEO database. By integrating the results from the various analyses, this study investigated the biological processes, pathways, and cell types associated with TBI and explored the activity of these cells during various TBI phases. The results showed the response to cytokine, inflammatory response, bacteria-associated response, metabolic and biosynthetic processes, and pathways of neurodegeneration to be involved in the pathogenesis of TBI. The cellular abundance of microglia, perivascular macrophages (PM), and neurons were found to differ after TBI and at different times postinjury. In conclusion, immune- and inflammation-related pathways, as well as pathways of neurodegeneration, are closely related to TBI. Microglia, PM, and neurons are thought to play roles in TBI with different activities that vary by phase of TBI.

## 1. Introduction

Traumatic brain injury (TBI) is common worldwide. Over 27 million traumatic brain injuries occur globally each year [1]. TBI is responsible for 30–40% of all injury-related deaths, and is by far the leading cause of disability associated with neurological diseases, accounting for 2–3 times more disabilities than Alzheimer’s disease or cerebrovascular disease [2]. There is even evidence that TBIs are associated with dementia years later [3]. Survivors endure enormous psychological, physical, and emotional pain, while their families and societies face enormous burdens. Although various therapeutic attempts have been made to improve the outcome of TBI, most multicentral clinical trials of medical and surgical interventions have failed to show efficacy [2]. For this reason, it is important to gain a holistic understanding of the mechanisms underlying TBI to come up with optimal treatment options.

TBI is not a single pathophysiological event but a complex disease process. For understanding the primary and secondary injury mechanisms in TBI, a number of preclinical animal models have been developed. Rat is one of the most widely used animals in TBI research due to its modest cost, small size, and standardized outcome measurements [4]. Through the research based on experimental models, many pathophysiological processes of TBI have been better understood, including disturbance in neurotransmitters and calcium signaling pathways [5]; abnormal lipids, proteins, and nucleic acids oxidation [6]; upregulation of transcription factors and inflammatory mediators [7]; and increased expression of detrimental cytokines, which induce brain edema, blood–brain barrier damage, and cell death [8].

With the advances in microarray and high-throughput sequencing techniques, large and growing public databases of TBI gene expression data are being deposited into public databases. The transcriptome analysis of gene expression has been used to identify pathways potentially involved in TBI. In most cases, gene expression profile data are analyzed by focusing on genes that differ between TBI and control groups while ignoring other genes that may also associated with sample features. Weighted gene coexpression network analysis (WGCNA) is a bioinformatics algorithm used for exploring gene association patterns in samples, identifying gene sets with highly coordinated expression, and exploring biologically meaningful gene sets connected to a particular trait [9]. In this study, transcriptional profiling data of rat brain tissues after TBI were analyzed using WGCNA. Feature-relevant modules containing coexpressed genes were identified, and their gene ontology (GO) functions and signaling pathways that may be involved were investigated. These gene sets were further subjected to gene set enrichment analysis (GSEA) to recognize cell types associated with the sample traits, which was further validated using gene set variation analysis (GSVA) by independent datasets. The results may provide a reference for the mechanism research and treatment of TBI.

## 2. Materials and Methods

### 2.1. Data Acquisition and Preprocessing

The Gene Expression Omnibus (GEO) database (accessed on 25 October 2021 from https://www.ncbi.nlm.nih.gov/geo/) was searched using the following terms: traumatic brain injury, TBI, brain trauma, or neurotrauma. A further filter was performed with the organism “Rattus norvegicus” and the study type “Expression profiling by array” or “Expression profiling by high throughput sequencing”. Search results were manually checked, datasets without sham-injury-treated control samples were excluded, and 14 datasets containing rat brain tissue samples were included in this study. Additionally, the single-nuclei sequencing dataset GSE137869 [10] was retrieved for cell marker identification. Among the datasets, GSE2871 [11] was used for WGCNA; GSE2392 [12], GSE2871 [11], and GSE45997 [13] were used in the GSEA analysis; GSE1911 [14], GSE2392 [12], GSE24047 [15,16], GSE31357 [17], GSE59645 [18], GSE64978 [19], GSE67836 [20], GSE68207 [19], GSE80174 [21], GSE86579 [22], GSE111452 [23], and GSE115614 [18] were analyzed for validation. Among the samples in the above datasets, only samples from wild-type rats with no additional treatment or disease other than TBI were included. Brain tissues from young rats in GSE137869 were used for single-nucleus transcriptome analysis. The sample traits were determined based on groupings and sample information in the database. If applicable, background correction and normalization were conducted using the R package limma (version 3.46.0) [24]. Detailed information is shown in Table 1.

### 2.2. WGCNA

To identify the gene modules relevant to TBI, the R package WGCNA (version 1.70) [25] was used to conduct the weighted co-expression network analysis. A power of 6, which enabled the scale-free topology fit index to reach 0.85, was selected as soft-threshold parameters to construct a signed, scale-free coexpression gene network. Thereafter, modules of coexpressed genes were identified by hierarchical clustering, and the minimum size of modules was set to 40 genes. Furthermore, the module eigengene (ME) representing each module’s expression profiles was calculated, and intramodular correlations and module–trait associations were estimated. Modules with high module–trait significance (*p*-value < 0.01) were defined as key modules and subjected to further analysis. The result of intramodular correlations and module–trait relationship analyses were plotted using the R package ggcor (version 0.9.8).

### 2.3. GO and KEGG Enrichment Analysis

Gene Ontology (GO) enrichment analysis provides a structured description of the known biological information of genes at different levels: biological process (BP) refers to a biological objective to which the gene or gene product contributes, cellular component (CC) refers to the place in the cell where a gene product is active, and molecular function (MF) is defined as the biochemical activity of a gene product [26]. Kyoto Encyclopedia of Genes and Genomes (KEGG) analysis assigns functional meanings to genes and genomes both at the molecular and higher levels [27]. GO enrichment analysis and KEGG analysis were performed for understanding the biological functions and pathways involved in genes in key modules. Both enrichment analyses were implemented in the R package clusterProfiler (version 3.18.1) [28]. A *p*-value < 0.05 and a q-value < 0.05 was considered significant. In each analysis, the top 10 results were extracted for visualization.

### 2.4. Computational Analysis of snRNA Seq Datasets

The single-nucleus transcriptome analysis was performed in the R package Seurat (version 4.1.0) [29,30,31,32]. The quality control process was as follows. Nuclei containing more than 2000 expressed genes or those that contained less than 200 expressed genes were removed. Data on nuclei that contained more than 2.5% mitochondrial genes were filtered. Features expressed in three or fewer nuclei were excluded. After logarithmical normalization of the filtered nuclei data, principal component (PC) analysis was performed. The first 15 PCs, determined using a combination of jackstraw and elbow methods, were used to generate clusters with a resolution of 1.5. For visualization, the nonlinear dimensional reduction was performed with the t-distributed stochastic neighbor embedding (t-SNE) algorithm. Marker genes for the individual clusters were identified using the FindAllMarkers function with default parameters. Annotation of cell types was manually conducted according to a previous study [10], and clusters identified as the same cell type were merged.

### 2.5. Gene Set Enrichment Analysis

To identify the key cell types associated with sample traits in TBI, gene set enrichment analysis (GSEA) was conducted. First, log2FC values representing the expression change between the compared samples were calculated using the R package limma (version 3.46.0). Then, the key modules related to the same sample trait from WGCNA were merged, and the genes contained in the modules were ranked according to log2FC. The rank list was used as input to GSEA, which was performed based on the cell type marker genes that could be detected and annotated in GSE2871 using the R package clusterProfiler (version 3.18.1). A *p*-value < 0.05 and a q-value < 0.25 was considered significant.

### 2.6. Gene Set Variation Analysis

Gene set variation analysis (GSVA) is a gene set enrichment method that estimates changes in gene set enrichment over the samples independently of any class label, and has emerged as an overall top method to assign cell type labels in single-cell RNA-sequencing analysis [33,34]. Expression statistics of the cell type markers are summarized into a single enrichment score for each cell type. GSVA was employed to estimate the abundance or activity of GSEA-enriched cell types in the validation datasets. R package GSVA (version 1.38.2) [26] was used to score individual samples based on the top 20 markers selected based on the likelihood-ratio of microglia, PM, and neurons. Enrichment scores were compared between differently treated samples within the same database. To aggregate enrichment scores from different datasets and make comprehensive comparisons within a wider range of time, the enrichment scores were normalized to the corresponding sham group as the relative enrichment score.

### 2.7. Statistical Analysis

Data analysis and plotting of the results were performed using R software (version 4.0.2) and GraphPad Prism (GraphPad Prism 8; GraphPad). Nonparametric test or *t*-test based on data distribution characteristics was used to distinguish the difference between the two groups, and a *p*-value of < 0.05 was considered significant.

## 3. Results

### 3.1. WGCNA Identified Key Modules Related to Sample Traits in TBI

Rat TBI dataset GSE2871 was downloaded and sample information was obtained from GEO. Briefly, adult rats were subjected to lateral fluid percussion injury (mild or severe) or sham surgery without injury. Expression profiling of brain regions (parietal cortex and hippocampus, ipsilateral and contralateral to injury) was conducted at 4 h or 24 h postinjury. All 8799 genes from 47 samples in GSE2871 were subjected to WGCNA. A scale-free network was constructed as described in Materials and Methods, and a total of 19 gene coexpression modules were obtained. Among the 19 modules, the turquoise module was the largest, which contained 2109 genes, while the light green module containing 63 genes was the smallest one. A total 930 ungrouped genes were included in the grey module (Figure 1A).

The correlations among modules and the association between modules and traits were estimated. Key modules in TBI were defined as modules with a high module–trait significance (*p*-value < 0.01). The modules associated with the severity of injury were the magenta module and the tan module, which contained a total of 265 annotated genes. The black, turquoise, magenta, and tan modules were linked to the sampling side (ipsilateral or contralateral to injury) and contained 2677 genes that were annotated. The green, yellow, brown, and tan modules, with a total of 2800 annotated genes, were associated with postinjury time. The green, brown, and blue modules were related to brain region (Figure 1B).

### 3.2. Function Enrichment Analysis of Key Modules

The sample traits “severity of injury”, “sampling side”, and “postinjury time” were considered as key features associated with TBI, and modules related to these sample traits were further analyzed. As part of our investigation of the biological functions of TBI-related genes, GO and KEGG enrichment analyses were performed on genes in key modules associated with each sample trait. The GO results showed that genes related to injury severity were mainly involved in response to cytokine, inflammatory response, and other immune-related processes (Figure 2A). Side-related genes largely played a role in the inflammatory response, bacteria-associated response, as well as involved in transmembrane signaling receptor activity (Figure 2B). Genes associated with postinjury time were mainly related to the metabolic and biosynthetic processes, cell junction organization, etc. (Figure 2C).

The KEGG analysis indicated that injury-severity-related genes were mainly enriched in lipid and atherosclerosis, and various virus and parasitic infection pathways (Figure 2D). Genes related to the sampling side were primarily involved in neuroactive ligand–receptor interaction, calcium signaling pathway, chemical carcinogenesis-receptor activation, etc. (Figure 2E). Those genes associated with postinjury time were mainly involved in multiple neurodegeneration pathways (Figure 2F).

Although functional enrichment analyses yielded a variety of biological functions and pathways, it is worth noting that immune- and inflammation-related terms were commonly enriched in several analyses. An interesting question then is what role various cells, especially immune-related cells, play in TBI. Therefore, on the basis of the above results, we continued to explore the changes in TBI-related cellular activity.

### 3.3. Identification of the Markers of Rat Brain Cell Types

To identify cell types in rat brains and find the corresponding markers, we collected snRNA-seq data of two rat brain tissue samples without any treatments or disease from the dataset GSE137869 and conducted the computational analysis as described in Materials and Methods. A total of 8889 nuclei with 16,882 features were subject to the analysis, which yielded 22 cell clusters. Of the 22 clusters obtained, six clusters were annotated as neurons, five clusters were annotated as oligodendrocytes, three clusters were annotated as oligodendrocyte progenitor cells (OPC), two clusters were annotated as microglia, and two clusters were annotated as astrocytes. The remaining four clusters were annotated as pericyte, endothelial cells (EC), perivascular macrophages (PM), and vascular leptomeningeal cells (VLMC), respectively. Clusters of the same cell type were merged. For each cell type, gene markers were identified for subsequent analysis (Figure 3). The analysis resulted in 148 marker genes for astrocytes, 98 marker genes for EC, 92 marker genes for microglia, 592 genes for neurons, 10 marker genes for oligodendrocytes, 229 marker genes for OPC, 94 genes for pericytes, 139 genes for PM, and 152 genes for VLMC.

### 3.4. Characterization of Key Cell Types Associated with Traits

To explore changes in cellular abundance and activity related to a sample trait, we performed a GSEA analysis of genes in key modules related to that trait using markers of cell types. Briefly, differential expression analysis was conducted on expression data from samples for the corresponding traits, and GSEA was performed on a gene list presorted by the fold-change value of the differential expression analysis. The enriched cell types were considered to be associated with the corresponding sample traits. Differential expression analysis was conducted between samples from the lesioned side and samples from sham-treated animals in GSE45997 to identify cell types associated with the severity of injury. Samples from the ipsilateral and contralateral side of the injury in GSE45997 were compared to rank the gene list, which was input into GSEA to locate side-related cell types. Similarly, samples of 24 h postinjury and 4 h postinjury in GSE2392 were used to identify time-sensitive cell types. For the severity of injury and side of injury, the above analysis was repeated by collecting corresponding trait samples from GSE2871. However, no replication was performed for postinjury time because, in GSE2871, the number of ipsilateral samples at 24 h postinjury was insufficient (*N* = 2) for given injury severity.

PM was enriched in the side-associated modules in both GSEA, and microglia was enriched in one of the analyses. The above results indicate more PM and microglia on the ipsilateral side of injury (Figure 4A,C,E,G,H). As for the postinjury time, neuron was enriched with an acceptable significant level (*p*-value = 0.047 and q-value = 0.227), indicating possibly decreased neuron abundance or activity 24 h postinjury compared with that at 4 h (Figure 4B,D). For the key modules related to injury severity, either no terms were enriched, or the enriched cell types did not reach significance in GSEA (Figure 4F).

### 3.5. Validation of Cell Activity after Traumatic Brain Injury

To validate the change in abundance or activity of microglia, PM, and neurons, GSVA was performed on the other datasets containing TBI samples and corresponding sham controls collected at various postinjury times, including 30 min (GSE2392), 3 h (GSE1911 and GSE24047), 4 h (GSE2392 and GSE31357), 6 h (GSE24047), 8 h (GSE2392), 12 h (GSE24047), 24 h (GSE1911, GSE2392, GSE31357, GSE59645, GSE111452, and GSE115614), 48 h (GSE24047), 72 h (GSE2392), 1 week (GSE64978 and GSE68207), 2 weeks (GSE92363 and GSE111452), 3 weeks (GSE2392), 1 month (GSE67836), 3 months (GSE80174, GSE86579, and GSE111452), 6 months (GSE111452), and 1 year (GSE111452). Enrichment scores between TBI condition(s) and the corresponding control(s) in each dataset were compared and the results were plotted.

Results from some datasets showed decreased enrichment scores of microglia at 3 h, 4 h, 6 h, 12 h, and 1 year after TBI, while the results from some datasets suggested an increased abundance of microglia at 4 h, 24 h, 72 h, 1 week, 2 weeks, 1 month, 3 months, and 6 months (Figure 5). As for PM, there was no statistically significant difference in PM abundance within 6 h, except for one sample from GSE1911 that showed a decrease in PM enrichment score at 3 h postinjury. The enrichment scores of PM increased at 8 h, 24 h, 48 h, 72 h, 1 week, 2 weeks, 3 weeks, 3 months, and 6 months (Figure 6). At 24 h, 72 h, 2 weeks, 1 month, 3 months, and 6 months postinjury, the enrichment scores of neurons were lower in the injury-treated samples (Figure 7).

However, it is worth noting that some results from different datasets are inconsistent. For example, GSE2392 showed increased microglia enrichment 4 h after injury, while GSE31357 showed a decrease. Furthermore, several results from some datasets were statistically significant, while results from another dataset at the same postinjury time point did not reach significance.

In order to intuitively understand the changes, we divided the sample sets into five clusters according to the setting of the postinjury time of the samples in the database: 30 min to 12 h representing the hyperacute phase of TBI, 1 day to 3 days representing the acute phase, 1 week to 3 weeks representing the subacute phase, 1 month to 3 months representing the chronic phase, and longer than 3 months. Results from the same cluster were aggregated. The results showed that the abundance of microglia decreased from 30 min to 12 h after TBI and increased from 1 week to 3 months after injury. There were no differences in enrichment scores after 3 months (Figure 8A). For PM, there was no difference in cellular abundance up to 6 h postinjury, whereas the injured group had higher enrichment scores from 8 h to 3 months postinjury. Similar to microglia, enrichment scores did not differ between the injured and sham groups after 3 months (Figure 8B). The neuron changed at the same time as the PM, but in a different direction (Figure 8C).

## 4. Discussion

TBI is a worldwide public health concern without major therapeutic breakthroughs in the past decades [2]. Understanding the mechanisms that underlie TBI is critical for developing effective treatment options and improving the prognosis of this condition. The last decade has witnessed the rapid development of high-throughput transcriptome analysis. Public databases containing extensive TBI gene expression data enable comprehensive analysis of the specific effects of TBI on various cell functions and pathways.

To the best of our knowledge, this is the first TBI comprehensive study that utilizes WGCNA. WGCNA adopts a hierarchical clustering tree to classify all genes into several gene sets, namely, modules, according to the degree of coexpression. The correlation between modules and sample traits was estimated so that modules that are highly correlated with sample traits could be identified and gene functions of related modules could be further studied. WGCNA significantly reduces errors caused by multiple testing problems inherent in microarray data, while maximizing the use of all data, as it uses all gene expression data from samples—instead of focusing only on differentially expressed genes—to construct the scale-free weighted network [35]. Furthermore, the scale-free weighted network has a high degree of robustness, which means that the errors in individual genes will not affect the overall results.

In this study, we obtained key modules significantly associated with the traits of rat brain trauma samples. The sampling side reflects the differences of the local lesion tissue relative to unaffected sites in the same experimental animal, and the severity of injury reflects differences between experimental animals treated according to various severity levels (naive, sham injury, mild TBI, or severe TBI). The key modules related to the severity of injury and the sampling side partially overlap, which is not surprising since they may all be involved in pathophysiological processes directly related to TBI. Based on functional enrichment, both severity- and side-related genes were enriched in the inflammatory response and immune-related processes, indicating that inflammation and immune regulation are important post-TBI processes. This finding is supported by previous studies. Studies have found that TBI induced an inflammatory response in the central nervous system, which may cause acute secondary injury [7,8,36,37]. Evidence showed that the inflammatory response following a TBI does not only affect the focal zone but also disseminates to remote brain areas [38]. Furthermore, neuroinflammation after TBI was found to link to neurodegeneration [7]. The key modules related to postinjury time reflect the change between the hyperacute phase (4 h) and the acute phase (24 h). Ranked first in the KEGG analysis was the entry “pathways of neurodegeneration—multiple diseases”. Just as mentioned above, TBI has been proved a risk factor for neurodegenerative disorders, including Alzheimer’s disease and Parkinson’s disease [39]. Although the development of neurodegenerative disease is a long-term process, axonal damage and disruption of transport during the acute phase of TBI have been found to alter the molecular mechanisms of pathological protein formation, such as α-synuclein, amyloid-beta peptide, and hyperphosphorylated tau [40,41].

It is well-known that microglia and macrophages are important players in inflammatory and immune-related responses [42,43,44]. In the enrichment analysis of trait-related cell types, we found that microglia and PM were associated with the side of injury. The abundance of both microglia and PM in TBI was further validated using the other rat TBI datasets. This is consistent with previous studies, as substantial evidence has suggested that the alteration of microglia was involved in the acute immune and neuroinflammatory response following TBI [45]. For example, microglia have been shown to play a neuroprotective role in TBI [46,47]. There is also evidence that, as well as clearing debris from the brain, microglia may also be involved in maintaining the integrity of the glial barrier after brain injury [45].

We also found for the first time that PM were significantly associated with TBI as microglia. However, the two types of cells differed in their association with postinjury time, as we revealed that the activation of PM and microglia was different within 3 days after injury. PM are specialized macrophages residing in the perivascular space of the brain. Similar to microglia, PM migrated from the yolk sac into the brain during embryonic development. As a result, they are likely to be a self-renewing cell population that is not replenished by circulating monocytes under a normal state [48]. Although PM have been implicated in various diseases, for example, the clearance of amyloid-beta in animal models of amyloid-beta pathologies, it is unclear what role they play in TBI [48,49,50,51].

However, care needs to be taken when interpreting these results, since no single marker has so far been able to reliably define PM with both good sensitivity and specificity [51] and monocytes are capable of crossing the blood–brain barrier into the injured brain [52]. In order to identify cell markers, we analyzed single-cell-nucleus sequencing data from normal rat brain tissues. In spite of using a relatively high resolution for the analysis, which yielded 22 cell clusters, only one cluster was annotated as macrophage, making it difficult to distinguish between monocyte-derived macrophages and PM. Contrary to the few studies on PM, a number of studies have investigated the role of peripherally derived macrophages in TBI. The number of monocyte-derived macrophages from the blood reaches a peak in the damaged brain of animals and humans 24 to 48 h after injury [53]. Studies have identified monocyte-derived macrophages as a pathogenic factor during the chronic phase of TBI [53,54]. However, macrophages from the peripheral circulation and macrophages residing in the brain are two distinct types of cells and, therefore, may have different immune reactions to TBI [45,55]. To better explore the cell types associated with TBI and their roles in TBI, more research needs to be conducted to establish definitive profiles of microglia and other CNS macrophages at different stages of TBI.

TBI can be divided into four phases: hyperacute (minutes to hours), acute (hours to several days), subacute (several days to weeks), and chronic (months and beyond) [56]. Our results showed that during the hyperacute phase (30 min to 12 h postinjury), the cellular activity of microglia in the injury group was significantly lower than in the sham group, while in the acute phase (1 day to 3 days postinjury) the difference was not significant. For PM, there was no difference between the two groups in the hyperacute phase. The cellular abundance was significantly increased starting from the acute phase. According to the differences in these two cellular activation patterns, it is possible that each cell type responds to a specific pattern of stimulation at each time point after injury, and they likely play different roles in TBI. Further studies are needed to determine the specific mode and mechanisms of microglia and PM activation. Our results also showed that both of these cell types were significantly more abundant than controls up to 3 months after injury, suggesting that they may play a long-term role in TBI. A deeper investigation of these two types of cells, especially the PM, could better facilitate the development of inflammation-targeted therapies to improve TBI prognosis. This study can provide a reference for the setting of postinjury time for subsequent studies on these two types of cells.

Despite being the first study to perform WGCNA analysis on TBI samples, and identifying PM as one of the relevant cell types associated with the sampling side and postinjury time, it has several limitations. The GSVA Enrichment scores of samples from various datasets were aggregated to intuitively understand the changes in cell abundance during various TBI phases. However, TBI models, RNA microarray platforms, and brain regions from which samples were collected vary from dataset to dataset. As a result, care needs to be taken when interpreting the results. Although TBI in rat models may share some similarities with human TBI, there may be variations in the course of rat and human TBI. Furthermore, the situation of TBI in humans is much more complex than in animal models. Therefore, the results of this study should be interpreted with caution. Although this study predicted relevant pathways and cell types involved in TBI and explored the activity of these cells during various TBI phases, further in vitro and in vivo experiments are needed to validate these findings.

In summary, this study performed WGCNA on TBI samples of various severity, ipsilateral and contralateral to injury, sampled from different brain regions at different times postinjury; identified key coexpression modules associated with traits of interest; unveiled several biological processes, pathways, and cell types associated with TBI; and explored the activity of these cell types at various TBI phases. The results of the current study may provide a reference for further mechanism research and treatment of TBI.

## 5. Conclusions

Using WGCNA, our study revealed response to cytokine, inflammatory response, bacteria-associated response, neuroactive ligand–receptor interaction, metabolic and biosynthetic processes, and multiple pathways of neurodegeneration to be involved in the pathogenesis of TBI. Microglia, PM, and neurons were recognized to associate with TBI with different activities that vary by phase of TBI.

## Figures and Tables

**Figure 1 jcm-11-03437-f001:**
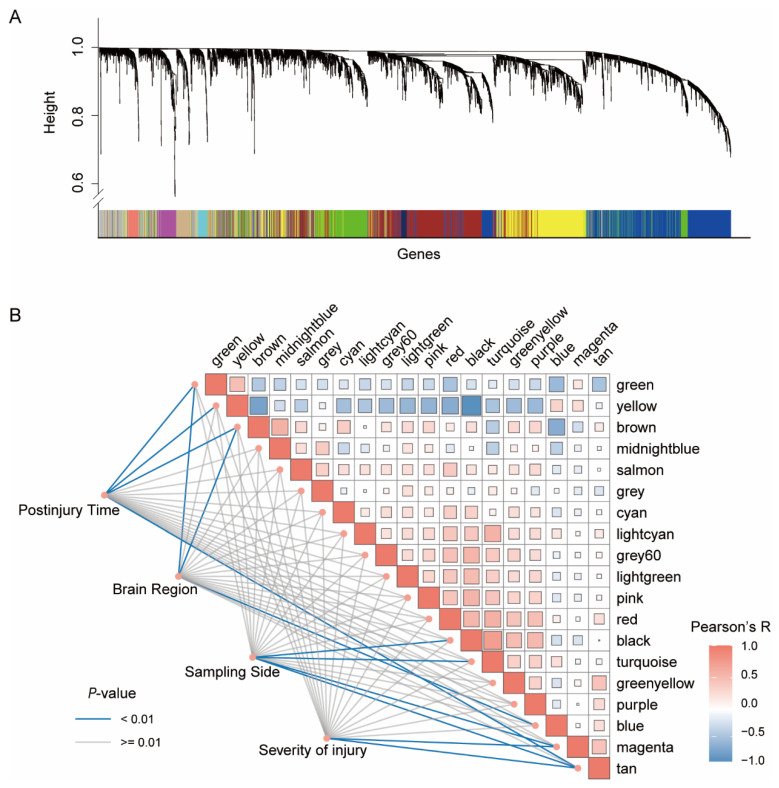
WGCNA of the rat TBI Dataset GSE2871. (**A**) Hierarchical clustering tree of genes, with dissimilarity based on the topological overlap. Each gene cluster (module) is marked with a different color. (**B**) Identification of key modules related to sample traits. Heatmap illustrating the intramodular relationship. Rows and columns correspond to modules, and each cell contains the corresponding correlation and *p*-value. Pearson’s R-values are color-coded according to the color legend. The size of the rectangle is proportional to the *p*-value; the larger the rectangle, the more significant the correlation. The color of the line represents the correlation significance between traits and modules.

**Figure 2 jcm-11-03437-f002:**
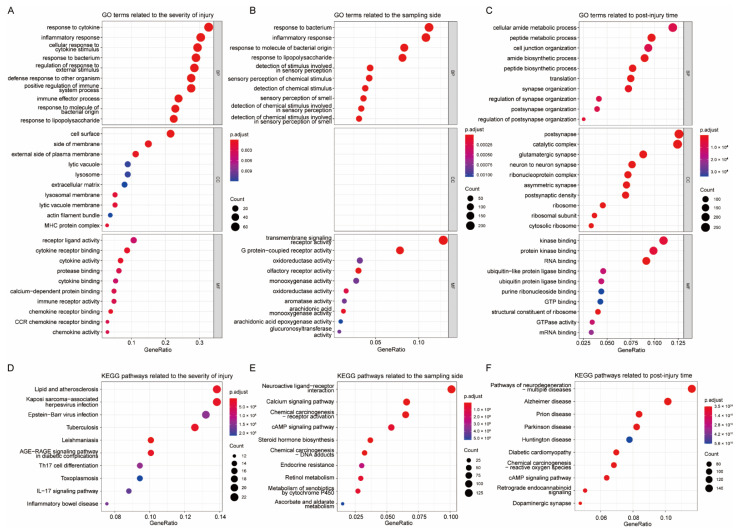
KEGG and GO enrichment analysis of the genes in sample-traits-related modules. (**A**–**C**) Dot plots illustrating GO terms enriched in modules related to the severity of injury (**A**), sampling side (**B**), or postinjury time (**C**); (**D**–**F**) dot plots showing KEGG pathways enriched in modules related to severity (**D**), side (**E**), and time (**F**). BP—biological processes, CC—cellular components, MF—molecular functions. Note: No term was enriched for CC in (**B**).

**Figure 3 jcm-11-03437-f003:**
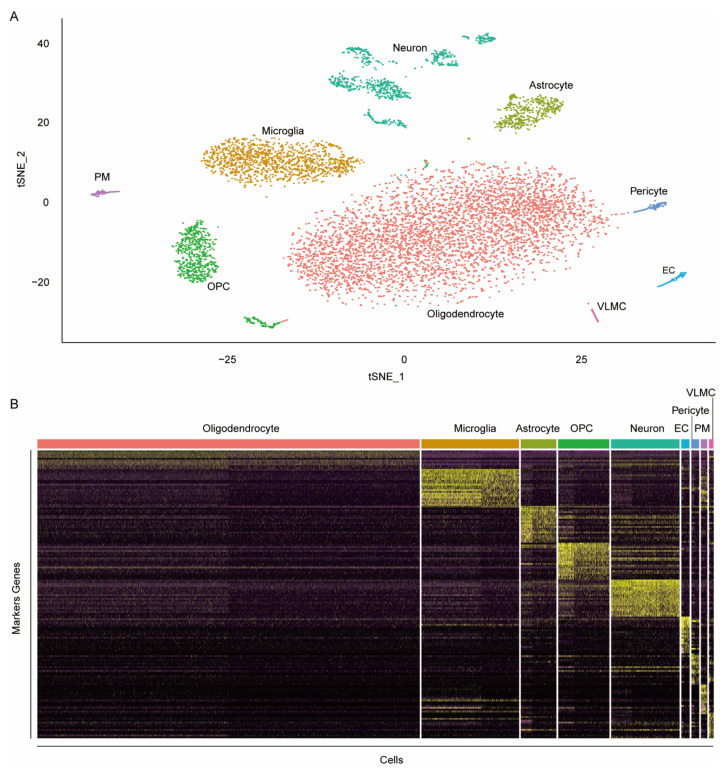
Single-nucleus transcriptome analysis. (**A**) The t-SNE plot of rat brain single-nucleus transcriptomes; (**B**) heatmap illustrating the expression levels of top 20 marker genes (rows) sorted by likelihood ratio for each cell type in each cell (columns), except for oligodendrocytes, which have only 10 genes.

**Figure 4 jcm-11-03437-f004:**
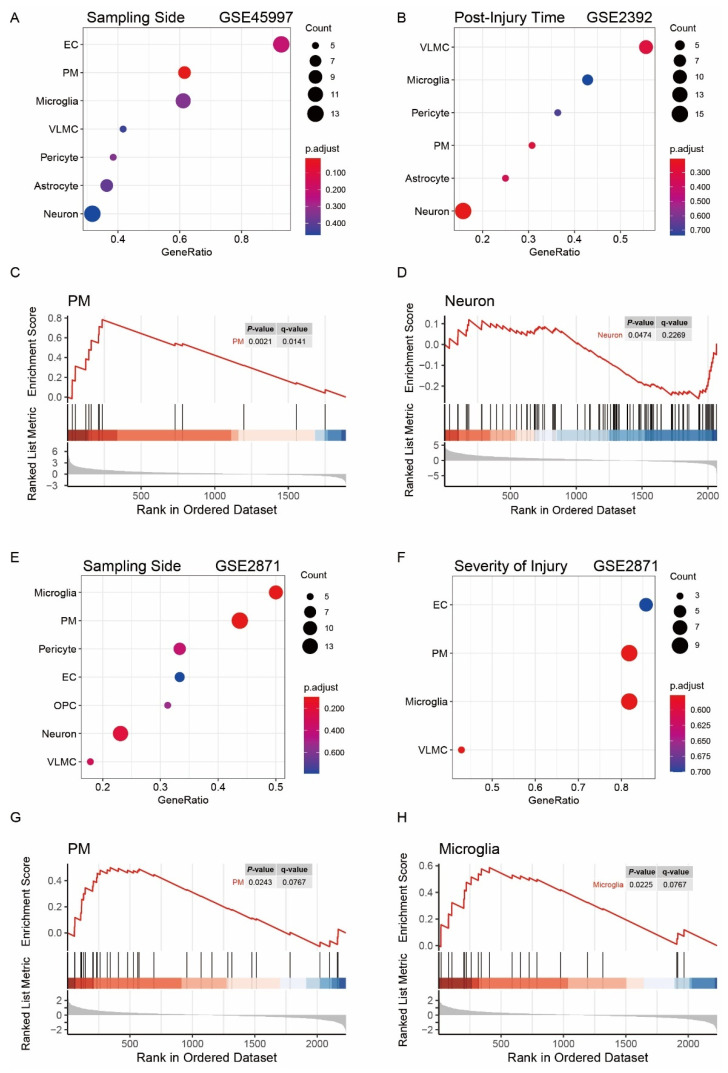
Gene set enrichment analysis (GSEA) of the genes in modules related to sample traits. (**A**,**C**) Dot plots illustrating cell types enriched in the genes in side-related modules using data from GSE45997 and the enrichment plots of PM; (**B**,**D**) dot plots illustrating cell types enriched in time-related modules using data from GSE2392 and the enrichment plots of neurons; (**E**,**G**,**H**) dot plots of the enrichments in side-related modules using data from GSE2871 and the enrichment plots of PM and microglia; (**F**) dot plots illustrating enriched cells in severity-related modules using data from GSE2871.

**Figure 5 jcm-11-03437-f005:**
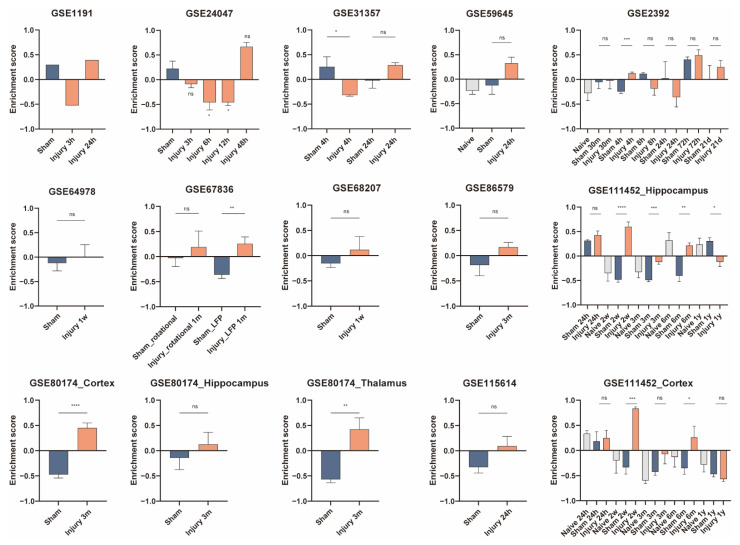
Gene set variation analysis (GSVA) for microglia in the validation datasets. GSVA enrichment scores from the validation data sets are plotted as bar graphs. The GEO accession number of the dataset is labeled in each panel. Samples from multiple regions in one dataset are compared separately. Information is labeled in the corresponding panels if the dataset contains samples with different treatments or harvested at various postinjury time points. For GSE1911, *N* = 1 for each group; for GSE24047, *N* = 4 for Sham group and *N* = 3 for the other groups; for GSE31357, *N* = 4 for each group; for GSE59645, *N* = 2 for Sham group and *N* = 3 for Naïve group and Injury 24-h group; for GSE2392, *N* = 3 for Naïve, Injury 4-h, Injury 8-h, Injury 24-h, Injury 72-h, and Injury 21-d groups, *N* = 4 for Sham 30-m and Sham 4-h groups, *N* = 5 for Injury 30-m group, *N* = 2 for Sham 8-h, Sham 24-h, Sham 72-h, and Sham 21-d groups; for GSE64798, *N* = 5 for each group; for GSE67836, *N* = 4 for Sham_LFP group and *N* = 3 for the other groups; for GSE68207, *N* = 4 for each group; for GSE86579, *N* = 5 for Sham group and *N* = 6 for Injury 3-m group; for GSE111452_Hippocampus, *N* = 2 for Sham 24-h group, *N* = 3 for Injury 24-h group, and *N* = 4 for the other groups; for GSE80174_Cortex, GSE80174_Hippocampus, and GSE80174_Thalamus, *N* = 5 for each group; for GSE115614, *N* = 2 for Sham group and *N* = 3 for Injury 24-h group; for GSE111452_Cortex, *N* = 4 for each group. **** *p* < 0.0001, *** *p* < 0.001, ** *p* < 0.01, * *p* < 0.05. Data are shown as means ± SEM.

**Figure 6 jcm-11-03437-f006:**
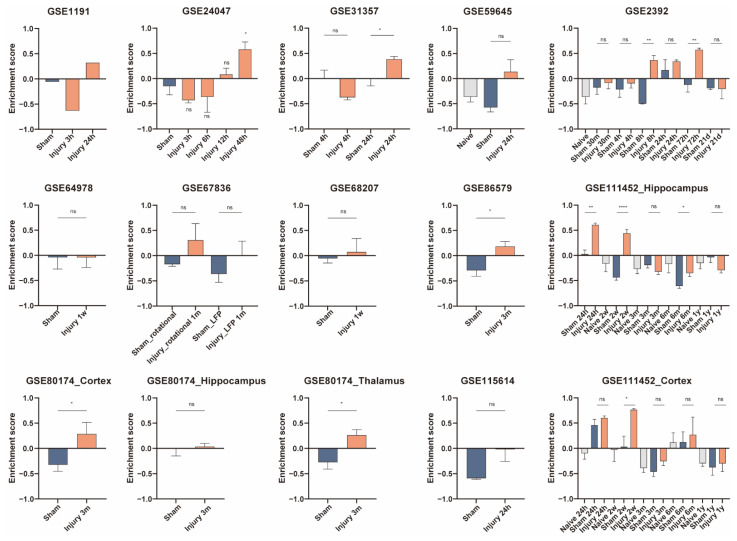
Gene set variation analysis (GSVA) for PM in the validation datasets. GSVA enrichment scores from the validation data sets are plotted as bar graphs. The GEO accession number of the dataset is labeled in each panel. Samples from multiple regions in one dataset are compared separately. Information is labeled in the corresponding panels if the dataset contains samples with different treatments or harvested at various postinjury time points. For GSE1911, *N* = 1 for each group; for GSE24047, *N* = 4 for Sham group and *N* = 3 for the other groups; for GSE31357, *N* = 4 for each group; for GSE59645, *N* = 2 for Sham group and *N* = 3 for Naïve group and Injury 24-h group; for GSE2392, *N* = 3 for Naïve, Injury 4-h, Injury 8-h, Injury 24-h, Injury 72-h, and Injury 21-d groups, *N* = 4 for Sham 30-m and Sham 4-h groups, *N* = 5 for Injury 30-m group, *N* = 2 for Sham 8-h, Sham 24-h, Sham 72-h, and Sham 21-d groups; for GSE64798, *N* = 5 for each group; for GSE67836, *N* = 4 for Sham_LFP group and *N* = 3 for the other groups; for GSE68207, *N* = 4 for each group; for GSE86579, *N* = 5 for Sham group and *N* = 6 for Injury 3-m group; for GSE111452_Hippocampus, *N* = 2 for Sham 24-h group, *N* = 3 for Injury 24-h group, and *N* = 4 for the other groups; For GSE80174_Cortex, GSE80174_Hippocampus, and GSE80174_Thalamus, *N* = 5 for each group; for GSE115614, *N* = 2 for Sham group and *N* = 3 for Injury 24-h group; for GSE111452_Cortex, *N* = 4 for each group. **** *p* < 0.0001, ** *p* < 0.01, * *p* < 0.05. Data are shown as means ± SEM.

**Figure 7 jcm-11-03437-f007:**
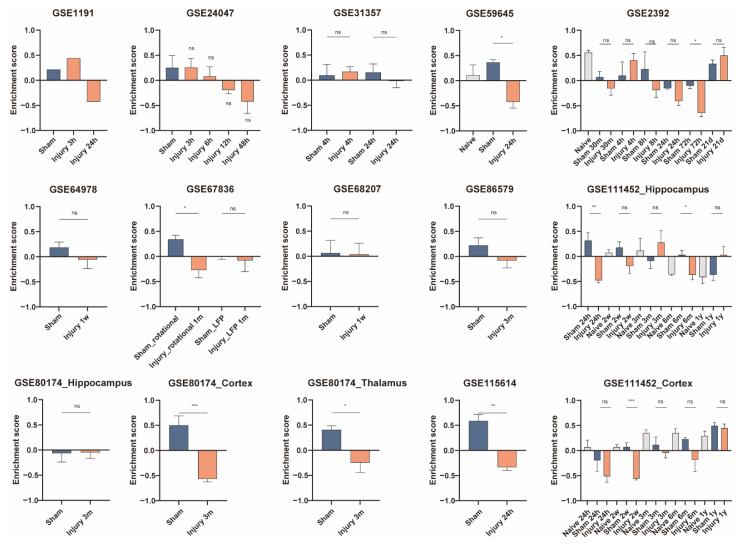
Gene set variation analysis (GSVA) for neuron in the validation datasets. GSVA enrichment scores from the validation data sets are plotted as bar graphs. The GEO accession number of the dataset is labeled in each panel. Samples from multiple regions in one dataset are compared separately. Information is labeled in the corresponding panels if the dataset contains samples with different treatments or harvested at various postinjury time points. For GSE1911, *N* = 1 for each group; for GSE24047, *N* = 4 for Sham group and *N* = 3 for the other groups; for GSE31357, *N* = 4 for each group; for GSE59645, *N* = 2 for Sham group and *N* = 3 for Naïve group and Injury 24-h group; for GSE2392, *N* = 3 for Naïve, Injury 4-h, Injury 8-h, Injury 24-h, Injury 72-h, and Injury 21-d groups, *N* = 4 for Sham 30-m and Sham 4-h groups, *N* = 5 for Injury 30-m group, *N* = 2 for Sham 8-h, Sham 24-h, Sham 72-h, and Sham 21-d groups; for GSE64798, *N* = 5 for each group; for GSE67836, *N* = 4 for Sham_LFP group and *N* = 3 for the other groups; for GSE68207, *N* = 4 for each group; for GSE86579, *N* = 5 for Sham group and *N* = 6 for Injury 3-m group; for GSE111452_Hippocampus, *N* = 2 for Sham 24-h group, *N* = 3 for Injury 24-h group, and *N* = 4 for the other groups; for GSE80174_Cortex, GSE80174_Hippocampus, and GSE80174_Thalamus, *N* = 5 for each group; for GSE115614, *N* = 2 for Sham group and *N* = 3 for Injury 24-h group; for GSE111452_Cortex, *N* = 4 for each group. *** *p* < 0.001, ** *p* < 0.01, * *p* < 0.05. Data are shown as means ± SEM.

**Figure 8 jcm-11-03437-f008:**
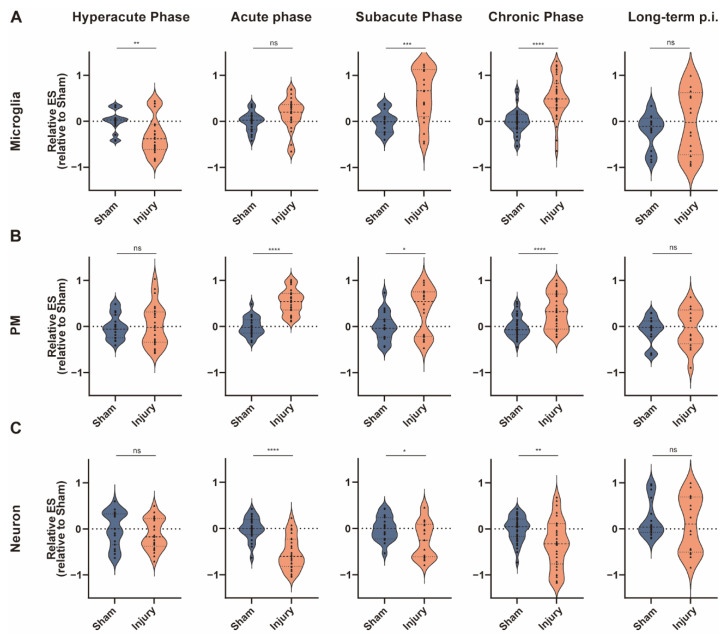
Analysis of cellular abundance of microglia, PM, and neurons during various courses of TBI. Violin plots showing aggregated GSVA enrichment scores for microglia (**A**), PM (**B**), and neurons (**C**) in hyperacute, acute, subacute, and chronic phases of TBI. For hyperacute phase, *N* = 27 for Sham group and *N* = 25 for Injury group; for acute phase, *N* = 23 for Sham group and *N* = 27 for Injury group; for subacute phase, *N* = 19 for Sham group and *N* = 20 for Injury group; for chronic phase, *N* = 35 for each group; for long-term p.i., *N* = 16 for each group. **** *p* < 0.0001, *** *p* < 0.001, ** *p* < 0.01, * *p* < 0.05. Data are shown as means ± SEM.

**Table 1 jcm-11-03437-t001:** GSE databases included in the study.

Dataset ID	TBI	Time	Tissue	Sample Number Included
GSE1911 [14]	CCI	3 h, 24 h	hippocampus	3
GSE2392 [12]	Moderate FPI	30 min, 4 h, 8 h, 24 h, 3 d, 3 w	perilesional cortex	39
GSE2871 [11]	Mild and severe FPI	4 h, 24 h	parietal cortex and hippocampus, ipsilateral and contralateral	47
GSE24047 [15,16]	FPI	3 h, 6 h, 12 h, 48 h	lateral cortex	16
GSE31357 [17]	TBI	4 h, 24 h	hippocampus	16
GSE45997 [13]	CCI	24 h	ipsilateral and contralateral brain	9
GSE59645 [18]	TBI	24 h	hippocampus	8
GSE64978 [19]	FPI	1 w	hippocampus	10
GSE67836 [20]	Rot-TBI and FPI	1 m	frontal cortex	13
GSE68207 [19]	FPI	1 w	Hippocampus	8
GSE80174 [21]	TBI	3 m	perilesional cortex, dorsal hippocampus, ipsilateral thalamus	30
GSE86579 [22]	FPI	3 m	hippocampus	11
GSE111452 [23]	FPI	24 h, 2 w, 3 m, 6 m, 1 y	hippocampus, cortex	113
GSE115614 [18]	TBI	24 h	hippocampus	5
GSE137869 [10]	-	-	brain	2

Abbreviations: CCI—controlled cortical impact; FPI—fluid percussion injury; Rot-TBI—rotational acceleration induced TBI; min—minute; h—hour; d—day; w—week; m—month; y—year.

## Data Availability

Datasets analyzed in the study can be accessed from Gene Expression Omnibus (GEO) database (https://www.ncbi.nlm.nih.gov/geo/ accessed on 25 October 2021) with the corresponding accession numbers listed in Materials and Methods.

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
