# Peer review of "Comprehensive RNA Expression Analysis Revealed Biological Functions of Key Gene Sets and Identified Disease-Associated Cell Types Involved in Rat Traumatic Brain Injury"

_jcm, 2022, doi:10.3390/jcm11123437_

Round 1

Reviewer 1 Report

Thank you for your work in TBI with respect to RNA expression analysis. This research showed differential RNA expressions by severity and time elapsing after trauma, and also changes in their expressions of each cell group in the brains of rat TBI models.

Followings are recommendations for amendments.

Abstract:

Line 25: closed à closely

Introduction:

Line 32: Traumatic brain injury à TBI

Line 35: traumatic brain injuries à TBIs

Materials and Methods:

Table 1 should include references (using reference numbers).

Line 76: The references should be given.

Line 77: Title of GSE137869 is ‘Caloric restriction reprograms the single-cell transcriptional landscape of Rattus norvegicus aging’. Why the authors did use aged mice? Since it does not seem to be normal or TBI rat, please explain the reason of using this set.

Lines 78-80: Is it right to use GSVA for validation analyses? Please specify the methods.

Lines 102-108: There is no explanation for each part analysis of GEO and KEGG enrichment analyses. Also, there is no supporting information for BP, CC, and MF in Fig.2. (for example, BP: biological precess)

Line 134: Please describe the background or rationale of selecting microglia and PM with a reference.

Please describe determination process about severity of TBI.
Please add explanation about “enrichment score” that was firstly mentioned in Lines 136-137.

Results

Fig. 2: Is there no result for CC in Fig 2 B? Or omitted?

Fig. 3 B: Please specify with box to differentiate column and row. There is no mention about meaning of the rows. Please list-up the top 20 markers.

Fig. 5, 6, 7: Would it be possible to supply information about time after injury in GESR59645, GSE64798, GSE67936, GSE68207,… which do not include the time? Also, please give n in each database.

Line 151: four à 4, to unify expression of numbers in this article.

Lines 160-165: There is no mention about brain regions in Fig. 1. The results of sampling side only described ipsi- or contralateral. Please add mention about cortex and hippocampus.

Lines 167-180: Also lacked results of brain region as above mention.

Lines 204, 207-210: Please unify expressions of numbers.

GO and KEGG enrichment analyses through WGCNA did not present co-expression network analysis result. The authors suggested possibility of network analysis by WGCNA, however, the results were given for expressions in each group (or cluster) and not inter-group relationship. It would be better if the result supply inter-group relationship in terms of pathway or mechanism.

Lines 234, 238: Please unify expression - 24 hours post-injury or 24 h post-injury

Line 234, lines 260-266, 302-313: Please unify expressions of numbers.

Discussion

Lines 423-431: While the Fig. 5-7 and Fig. 8 revealed differential responses after TBI according to time lapse, each dataset included different brain regions and symptom of TBI. And this could be a limiting condition for interpretation. It would be better to add further analyses or comment on these results.

There is no comment on astrocyte which has been known to be closely related in neuronal activity in TBI. Is there a reason, or no significant findings about it?

Reviewer 2 Report

Very specific, hard to read sometimes.

Please rewrite the hard to read paragrafs, keep în mind to ensure that supraspecialized terms are expleined.

Author Response

We would like to thank the reviewer for the comments and suggestions! We have adjusted the descriptions and expleinations in the manuscript for better clarity.

Round 2

Reviewer 1 Report

All issues were solved. Thank you for your work.